# Are Human Judgments of Real and Fake Faces Quantum-like Contextual?

**DOI:** 10.3390/e27080868

**Published:** 2025-08-15

**Authors:** Peter Bruza, Aaron Lee, Pamela Hoyte

**Affiliations:** Faculty of Science, Queensland University of Technology, Brisbane 4000, Australiapamela.hoyte@qut.edu.au (P.H.)

**Keywords:** quantum cognition, contextuality, AI

## Abstract

This paper describes a crowdsourced experiment in which participants were asked to judge which of two simultaneously presented facial images (one real, one AI-generated) was fake. With the growing presence of synthetic imagery in digital environments, cognitive systems must adapt to novel and often deceptive visual stimuli. Recent developments in cognitive science propose that some mental processes may exhibit quantum-like characteristics, particularly in their context sensitivity. Drawing on Tezzin’s “generalized fair coin” model, this study applied Contextuality-by-Default (CbD) theory to investigate whether human judgments of human faces exhibit quantum-like contextuality. Across 20 trials, each treated as a “generalized coin”, bootstrap resampling (10,000 iterations per coin) revealed that nine trials demonstrated quantum-like contextuality. Notably, Coin 4 exhibited strong context-sensitive causal asymmetry, where both the real and synthetic faces elicited inverse judgments due to their unusually strong resemblance to one another. These results support the growing evidence that cognitive judgments are sometimes quantum-like contextual, suggesting that adopting comparative strategies, such as evaluating unfamiliar faces alongside known-real exemplars, may enhance accuracy in detecting synthetic images. Such pairwise methods align with the strengths of human perception and may inform future interventions, user interfaces, or educational tools aimed at improving visual judgment under uncertainty.

## 1. Introduction

The proliferation and sophistication of AI-generated imagery, particularly deepfakes, has introduced complex challenges to human perception in the digital age. As these synthetic images become increasingly indistinguishable from authentic ones, the ability to accurately discern between real and fake visual stimuli is poised to become a critical cognitive skill. Important for the broader implications for trust, media literacy, and perceptual integrity, this study investigates whether human judgments of face images as being real or fake exhibit quantum-like contextuality. Quantum-like contextuality refers to a form of context sensitivity that is seemingly recalcitrant to traditional causal explanations [1]. In this study, participants were presented with pairs of faces (one real and one AI-generated), and were asked to judge which was fake. We interpret the quantum-like contextuality of a facial judgment to mean that the judgement cannot be reduced to a causal explanation based on the traits of the image itself or of the one with which it is paired. Given the increasing sophistication of generative algorithms, we speculate that successful discrimination between real and fake faces may depend not only on perceptual acuity but also on deeper quantum-like cognitive processes.

One reason for this speculation is the assumption that features relevant to discrimination may not have pre-existing values, much like quantum properties that are indeterminate until measured. In this way, the decision to identify one face as fake may not reflect a stable preference but instead emerge from a contextually structured judgment process. This view is supported by the findings of Nightingale S.J. [2], who demonstrated that participants’ ability to distinguish images of faces as AI-synthesized or real averaged near chance. While this result has been attributed largely to the high realism of synthetic imagery, which makes it challenging to distinguish real from fake faces, we posit that judgments are not fully determined by evaluation of facial or image features, but emerge through a quantum-like interaction between the facial stimulus and context. In our study the context is another face. More specifically, in the experiment reported below, human participants are shown a pair of face images where one is real and one is fake, and they are asked to judge which image is fake. Given that the probability of being successful in their determination approximates that of a fair coin toss, the judgment problem can be modeled as a multifaceted coin, which has been shown to exhibit contextuality [3]. By applying the Contextuality-by-Default (CbD) framework, our study empirically tests whether human judgments of facial authenticity across systematically varied real-fake pairings exhibit quantum-like contextuality, suggesting that the act of perceptual judgment may follow principles more aligned with quantum cognition than classical decision theory. Understanding this transition from quantum-like contextual processing to explicit computational reasoning based on cognitive states that are ’classical’, in the sense that they are always in a definite state, is important for anticipating how humans will adapt to environments increasingly saturated with realistic synthetic representations.

## 2. Contextuality of a Generalized Coin Toss

In order to provide a model with which to explore the quantum-like contextuality of facial judgments, a generalized coin is used [3,4]. The core idea is that the generalized coin is a “black box” with inputs and outputs. An input *x* specifies what experiment is to be performed on the black box, which produces, for example, an output *a*. For example, when a normal two-faced coin is viewed as a black box, the experiment *x* is a “coin toss” and the output *a* is “heads”. The importance of this framework is that the “entire physics is encapsulated in P(a|x), the probability that output *a* occurs given that measurement *x* was made” [5].

A single coin has one pair of opposite sides, with the possible outcomes of “heads”, which is denoted by ⊤ or “tails”, denoted by ⊥. A generalized coin has n>1 pairs of opposite sides. For example, n=4 coin is illustrated in Figure 1. In this coin, there are four pairs of distinct colors, namely, red, green, blue and yellow. Each pair can be viewed as a distinct facet $A_{i},\;0\leq i<4$ of the coin. Each facet has opposite sides, namely, a dark or light version of the color, e.g., dark or light blue, which are the analogs of “heads” and “tails” of a normal coin. More specifically, when a facet is measured, the outcome that is observed is *either* a light *or* dark version of the color.

A coin flip of the generalized coin occurs in the following way: Just as a normal coin has a single axis of rotation when tossed, the generalized coin of Figure 1 has four axes of rotation, one axis associated with each color. In addition, there are detectors that record the results of the measurement. In regard to a normal coin, the detector records ⊤ (“heads”) or ⊥ (“tails”). Generalized coins offer the possibility of simultaneous measurements where the detectors can distinguish between light and dark colors. More specifically, there are four axes and four detectors, where each detector is associated with a single axis. A toss is a rotation around an axis with the restriction that only two adjacent colors can be simultaneously measured. Figure 2 illustrates a toss of a the generalized coin depicted in Figure 1. The axis of rotation is shown by the dotted line. There are two adjacent detectors, $A_{0}$ which can record light or dark blue, and $A_{1}$, that can detect light or dark yellow. The detectors constitute a measurement context denoted $\left\{A_{0},A_{1}\right\}$.

Due to there being four axes of rotation restricted to adjacent detectors, there are four measurement contexts of simultaneous measurements $\left\{A_{0},A_{1}\right\},\left\{A_{1},A_{2}\right\},\left\{A_{2},A_{3}\right\},\left\{A_{3},A_{0}\right\}$, which constitute a cyclic system of random variables over outcomes {⊤,⊥} [6]. As we will see later, this fact allows Contextuality-by-Default theory to be applied to examine the contextuality of the generalized coin.

Tezzin [3,4] found that the generalized coin exhibits contextuality even though it is a classical system. The essence of the argument is that the generalized coin violates classical realism because measurement of the coin does not record pre-existing values of properties of the system, e.g., “There is no mechanical observable being measured when a coin flip is performed; there is nothing like a “flipness” property for such a system” [3]. Tezzin states that the connection to contextuality is via the Kochen–Specker theorem which states that self-adjoint operators on a Hilbert space H,(dim(H)>2) cannot be viewed as representing properties of the system in such a way that these properties have definite values at any given time, which measurement simply “reads off”.

### 2.1. Formalisation of the Generalized Coin

Following [4], an n−cycle is a measurement scenario $S_{n}$ containing *n* dichotomous measurements $A_{0},\ldots ,A_{n-1}$ and *n* maximal measurement contexts $C_{i}\equiv \left\{A_{i},A_{i+1}\right\},\;i=0,\ldots ,n-1$ (if i=n−1, then i+1=0).

The generalized coin toss of Figure 2 is defined as follows:

**Definition** **1**(generalized coin toss)**.**
*Let $S_{4}$ be the 4-cycle. A “generalized coin toss” on $S_{4}$ is the behavior p such that, for any measurement context $C_{i}\equiv \left\{A_{i},A_{i+1}\right\}$, $p(\cdot |C_{i})\equiv$ $p(\cdot |A_{i},A_{i+1})$ is given by:*(1)$$\begin{array}{c} p\left(⊥,⊥|A_{i},A_{i+1}\right)=0=p\left(⊤,⊤|A_{i},A_{i+1}\right) \end{array}$$(2)$$\begin{array}{c} p\left(⊥,⊤|A_{i},A_{i+1}\right)=\frac{1}{2}=p\left(⊤,⊥|A_{i},A_{i+1}\right) \end{array}$$

In the preceding definition, each $A_{i}$ corresponds to a facet (or color) of a coin. A measurement of $A_{i}$ yields whether the facet is dark, denoted by ⊤, or light, denoted by ⊥. Therefore, the probability *p* associates a context-independent probability distribution $p(\cdot |A_{i})$ over the measurement outcomes {⊤,⊥}. In other words, their measurements are “nondisturbing” [4], which is a requirement for determinations of the contextuality of a system. In addition, all sides of the coin are fair: $p\left(⊥|A_{i}\right)=p\left(⊤|A_{i}\right)=\frac{1}{2}$. Finally, Equation (Equation 2) specifies that joint outcomes in a measurement context $C_{i}\equiv \left\{A_{i},A_{i+1}\right\}$ perfectly anti-correlate.

### 2.2. Analysing the Contextuality of a Multifaceted Coin

Tezzin [3] analyzes the contextuality of the multifaceted coin using exclusivity graphs. In the following, we apply another form of analysis based on probabilities, which opens the door to relate the contextuality of the judgments of real/fake faces.

George Boole’s ‘conditions of possible experience’ describe inequalities that relative frequencies must adhere to in order for the joint probability distribution to exist [7]. For example, if $A_{0},A_{1},A_{2}$ and $A_{3}$ are viewed as set-based events with associated frequency-based probabilities *p*, then



$\;p\left(A_{0}\cup A_{1}\cup A_{2}\cup A_{3}\right)=p\left(A_{0}\right)+p\left(A_{1}\right)+p\left(A_{2}\right)+p\left(A_{3}\right)-p\left(A_{0}\cap A_{1}\right)-p\left(A_{0}\cap \;A_{2}\right)-p\left(A_{0}\cap A_{3}\right)-p\left(A_{1}\cap A_{2}\right)-p\left(A_{1}\cap A_{3}\right)-p\left(A_{2}\cap A_{3}\right)+p\left(A_{0}\cap A_{1}\cap A_{2}\right)+p\left(A_{0}\cap \;A_{2}\cap A_{3}\right)+p\left(A_{1}\cap A_{2}\cap A_{3}\right)-p\left(A_{0}\cap A_{1}\cap A_{2}\cap A_{3}\right).$



Therefore,(3)$$\begin{array}{cc} p_{0}+p_{1}+p_{2}+p_{3}-p_{01}-p_{02}-p_{03}-p_{12}-p_{13}-p_{23} & \leq 1 \end{array}$$
where $p_{i}$ is shorthand for $p(A_{i})$ and $p_{ij}$ is shorthand for $p(A_{i}\cap A_{j})$. Referring back to Definition 1, let $p_{i}$ denote $p(⊤|A_{i})$ and $p_{ij}$ denote $p(⊤,⊤|A_{i},A_{i+1})$, then the n=4 generalized coin violates the inequality in Equation (Equation 3) as $p_{i}=0.5,\;0\leq i\leq 3$ and $p_{ij}=0$. Consequently, a joint probability distribution spanning $A_{i},\;0\leq i\leq 3$ does not exist, a result which is often taken as the signature of contextuality provided that there is no disturbance in the system [6,8]. The non-disturbance requirement entails that the marginal probabilities $p_{i}$ are stable across all joint measurements.

Pitowsky [7] states that when a “condition of possible experience” is violated, “maybe … there are no well-defined properties, existing independently of observation and distributed in a specific manner. All that exists are the phenomena themselves, which simply occur without cause” (p. 107). In other words, according to Pitowsky, the fact that the generalized coin violates a “condition of possible experience” can lead to the conjecture that colors of the coin do not exist prior to measurement, a conclusion that aligns with Tezzin’s previously mentioned Kochen–Specker-motivated designation of contextuality regarding the generalized coin. However, Pitowsky also states that the colors revealed by simultaneous measurement “simply occur” and cannot be reduced to a causal explanation. One way to understand this is due to nondisturbance, because disturbance is often assumed to be reducible to a causal explanation, e.g., the measurement outcome produced by one color detector is causally affecting the outcome of the other.

In contrast, investigations into the quantum-like contextuality in human cognition have often involved disturbance [9,10,11]. The Contextuality-by-Default (CbD) defines a necessary and sufficient condition to determine contextuality in the presence of disturbance [6,12].

The necessary and sufficient condition for determining contextuality on a cyclic system like $S_{n}$ is specified by Equation (Equation 4).(4)$$\begin{array}{cc} s_{\mathrm{odd}}-\left(n-2\right)-\Delta & >0 \end{array}$$
The quantity $s_{\mathrm{odd}}$ is the maximal summation of the correlations between pairs of variables in measurement contexts where the summation involves an odd number of minus signs. The value Δ measures the amount of disturbance in the system calculated by summing the absolute differences of the marginal probabilities across the four pairwise probability distributions.

As described above, the generalized coin depicted in Figure 1 involves a 4-cycle measurement scenario comprising the measurement contexts: $C_{0}\equiv \left\{A_{0},A_{1}\right\},\;C_{1}\equiv \left\{A_{1},A_{2}\right\},\;C_{2}\equiv \left\{A_{2},A_{3}\right\},\;C_{3}\equiv \left\{A_{3},A_{0}\right\}$. According to Definition 1, pairs of variables in each measurement context perfectly anti-correlate. Therefore, $s_{\mathrm{odd}}=3$. In addition, there is no disturbance (Δ=0) as the probability distributions $p(\cdot |A_{i})$ are context independent, with the marginal probabilities equal to 0.5 in all cases. Applying Equation (Equation 4)(5)$$\begin{array}{cc} s_{\mathrm{odd}}-\left(n-2\right)-\Delta & >0 \end{array}$$(6)$$\begin{array}{cc} 3-(4-2)-0 & >0 \end{array}$$(7)$$\begin{array}{cc} 1 & >0 \end{array}$$
which demonstrates that the inequality holds and the generalized coin depicted in Figure 1 is contextual as all marginal probabilities equal 0.5, i.e., Δ=0.

The preceding approach allows the empirical determination of whether a generalized coin is quantum-like contextual. The approach is distinct from empirical determinations of the quantum-like incompatibility of cognitive properties using the q−test to analyze question order effects ([11], ch5).

We will now describe an experiment where the design is based on an n=4 generalized coin where the sides are not light/dark colors but face images judged as fake/real.

## 3. Materials and Methods

The experiment is based on the task of judging which image is fake when presented with two faces side-by-side. In each image pair presented to participants, one is an AI-generated (fake) face and the other image is the face of a real person. In terms of the multifaceted coin, face images correspond to colors, and human judgments of fake (⊤) or real (⊥) correspond to the measurement outcomes of the ‘coin flip’.

### 3.1. Design and Materials

The experiment stimuli consisted of 80 images of white faces (40 real and 40 AI-generated), randomly selected from the set of faces studied in [2], and the number of male and female images were balanced in both the real images and the fake images. This subset was used as synthetic white faces were shown in their study to be the most challenging to accurately judge as being fake. From these, 20 sets of four faces were produced with each set consisting of two real and two fake images, and all four images within a set were either male or female, in order to mitigate gender bias. These are denoted $\left\{F_{0},R_{1},F_{2},R_{3}\right\}$ where $R_{i}$ denotes a real face, and $F_{j}$ denotes an AI-generated fake face.

Four measurement contexts were defined from each set of four faces as follows:$$\begin{array}{c} C_{0}\equiv \left\{F_{0},R_{1}\right\}\\ C_{1}\equiv \left\{R_{1},F_{2}\right\}\\ C_{2}\equiv \left\{F_{2},R_{3}\right\}\\ C_{3}\equiv \left\{R_{3},F_{0}\right\} \end{array}$$
A between-subjects design was used in which each participant is presented with twenty pairs of images within one of the given measurement contexts $\left(C_{0},C_{1},C_{2},C_{3}\right)$, with participants presented with ten pairs of female faces first, followed by ten pairs of male faces.

In each image pair, the order of presentation of real and fake faces was counter-balanced to account for order effects. That is, within each of the four experimental conditions, for any given face-pair, 50% of the participants were presented with the real face on the left and the fake on the right, and the other 50% of participants received the same faces but with the order reversed.

In short, the experimental design is derived from 20 generalized coins like the one depicted in Figure 3.

### 3.2. Participants

Two hundred participants were recruited from the crowd-sourcing platform Prolific, with 50 participants in each of the four measurement contexts. This number of participants is similar to the number of participants used by [2] in their experiments 2 and 3. Participants were over 18 years of age and from a variety of countries across North America, Europe, UK, Australasia, the Middle East and Asia. All participants had been verified as proficient in English by Prolific, and had an approval rate in previous studies above 95%. Remuneration was in the form of a small payment (£0.50), as per Prolific convention, and informed consent was obtained from all subjects involved in the study prior to commencement of the experiment.

### 3.3. Procedure

The participants were instructed that they would be presented with pairs of images, in which one of the images is real and the other is fake, and that they must judge which of them is fake. Each image pair is shown side-by-side and participants were asked to indicate which of them is fake by clicking on a radio button under the image (see Figure 4). Each participant was shown 20 image pairs—10 pairs of female faces, followed by 10 pairs of male faces, making 20 judgements. After the ten female face pairs were presented, participants were asked to provide short textual descriptions of how they were making their judgments. This was repeated after the 10 male face pairs were presented.

### 3.4. Analysis Methods

See Table 1, Table 2, Table 3 and Table 4 for the four probability distributions resulting from the four measurement contexts.

The measure of disturbance in the multifaceted coin flips is the sum of the absolute differences between the marginal probabilities:(8)$$\begin{array}{cc} \Delta & =\;|p_{0}-p_{3}\left|\;+\;\right|\left(1-p_{0}\right)-\left(1-p_{1}\right)\left|\;+\;\right|p_{1}-p_{2}\left|\;+\;\right|\left(1-p_{2}\right)-\left(1-p_{3}\right)| \end{array}$$
For example, the marginal probability, $p(⊤|F_{0})$ that face $F_{0}$ is judged fake in context $C_{0}$ is equal to $p\left(⊤,⊥|F_{0},R_{1}\right)+p\left(⊤,⊤|F_{0},R_{1}\right)=p_{0}$. Similarly, the marginal probability, $p(⊤|F_{0})$, that face $F_{0}$ is judged fake in context $C_{3}$ is equal to $p\left(⊥,⊤|R_{3},F_{0}\right)+p\left(⊤,⊤|R_{3},F_{0}\right)=p_{3}$. The absolute difference between these marginal probabilities is therefore computed by the term $|p_{0}-p_{3}|$ in Equation (Equation 8). The other terms in this equation express, respectively, the differences in marginal probabilities that $R_{1},\;F_{2}$, and $R_{3}$ are judged fake.

## 4. Results

Table 5 summarizes the results. Following Bruza et al. [1], Basieva et al. [10], Khrennikova [13], independent random resamples (10,000 iterations per coin) were computed for each coin to obtain a 95% confidence interval.

Consistent with Nightingale S.J. [2], we found that participant accuracy was approximately chance across all image types, for both female and male images, and across all images (see Table 6). However, none of the coins were found to approximate an ’ideal’ coin, where all the probabilities $p_{0},\ldots ,p_{3}$ must be 0.5.

Nine out of the 20 coins (i.e., image sets)—four of the female coins, and five of the male coins—were found to be CbD contextual, i.e., the CbD value both fell within the confidence interval, and the lower value of the confidence interval > 0. We note that for any of the un-bolded CbD values that the coin could be explained causally, because the level of disturbance is above the threshhold that Cbd theory allows for a determination of contextuality. Therefore, the higher the disturbance Δ, the more the results are amenable to a causal explanation. Notably, Coin 4 exhibited strong context-sensitive causal asymmetry, where both the real and synthetic faces elicited inverse judgments due to their unusually strong resemblance to one another. Conversely, for the CbD values found to be contextual, the judgements of the image as fake cannot be reduced to a causal explanation.

## 5. Discussion

Oosterhof and Todorov [14], Todorov et al. [15] have shown that judgments of whether a face is deemed trustworthy involve the determination of facial traits such as intelligence, attractiveness, and dominance. It is reasonable to assume that when participants are trying to determine whether a face is fake, they rely on the determination of certain traits that may influence that judgement [1]. A natural explanation is that the determination of these traits exerts a causal influence on the judgement that a given face *A* is fake. In addition, this judgment is made in the context of a face *B* that it is jointly presented with. Further, it is natural to assume that the traits established for *B* can causally influence the judgement regarding face *A*. This assumption was supported by qualitative data supplied by the participants, e.g., “Some looked very symmetrical so not real”, “generated ones just look ‘too perfect”’, “the hairline does not seem genuine”. Note that *A* appears in two measurement contexts; namely, in one measurement context *A* is jointly presented with face $B_{1}$, and in the second measurement context, *A* is presented together with face $B_{2}$. As $B_{1}$ and $B_{2}$ are distinct faces, it is reasonable to assume that their causal influence on the judgement whether *A* is fake will differ. A probabilistic consequence of the preceding argument is that there is disturbance, i.e., the marginal probability that *A* is fake will differ across the two measurement contexts, and that difference can be attributed to the varying causal influences exerted by face $B_{1}$ and face $B_{2}$. The preceding situation characterizes a situation in which context-sensitivity is reducible to a causal explanation. It is our view that non-contextual coins can be explained in this way.

In contrast, we speculate that where the coin is contextual, the judgements behave like a property, which is indeterminate before measurement, like the spin of a photon.

Moreover, contextuality entails the value of judgement would be different across measurement contexts for the same face. Consequently, *A* might be judged as fake when presented with $B_{1}$ and not judged as fake when presented with face $B_{2}$. Whilst the preceding constitutes context sensitivity, what is curious is that the level of disturbance is within the bounds that allow a determination of contextuality by CbD. Therefore, our view is the context-sensitivity does not have a causal basis. Our position follows that of Pitowsky given above, namely, “all that exists are the phenomena themselves, which simply occur without cause”. In this study, the phenomena are quantum-like cognitive properties. Another possibility, which we have not explored in this paper, is that the context-sensitivity of these contextual coins may be attributed to quantum causality [16].

Furthermore, the contextuality of the coin relates not only to the cognitive properties that do no have definite pre-existing values, but also the particular design of the experiment and the images used. In this sense, cognitive phenomena are like quantum phenomena — their contextuality cannot be considered independent of the experimental conditions in which they are studied.

## 6. Conclusions

Given the rapid rise of generated human images and deepfakes, the implications for humans interacting with AI-generated fake images warrants some speculation. This is especially so with the emergence of immersive technologies, such as virtual reality and augmented reality, where it is becoming increasingly difficult to differentiate actual reality from a synthetic one. We have put forward the view that both causally induced context-sensitivity and quantum-like contextuality (acausal context-sensitivity) are involved in the cognitive processes which make this differentiation in regard to pairs of real and synthetic faces that are presented together as pairs. In our view, quantum-like contextuality essentially means that the judgement that a face is fake is recalcitrant to a causal explanation based on the traits of the image itself or of the one with which it is paired.

A speculative future direction based on the findings of this study would involve adopting comparative strategies, such as evaluating unfamiliar faces alongside known-real exemplars, which may enhance accuracy in detecting synthetic images. Such pairwise methods align with the strengths of human perception and may help clarify when these perceptions are causally vs. acausally context-sensitive.

## Figures and Tables

**Figure 1 entropy-27-00868-f001:**
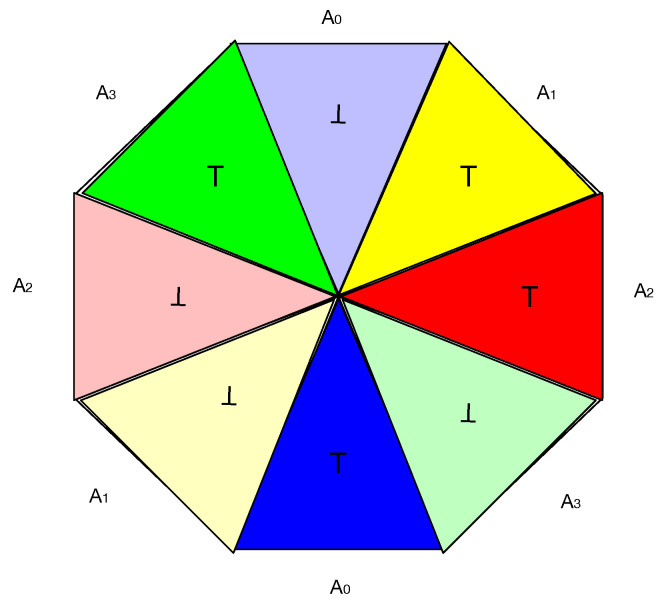
Schematic representation of a generalized coin with four colored facets.

**Figure 2 entropy-27-00868-f002:**
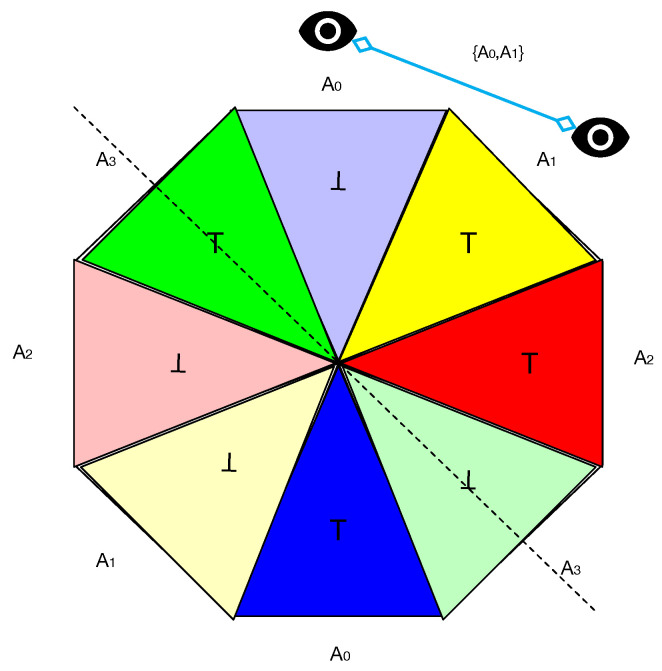
Coin toss of a generalized coin with four colored facets, with simultaneous measurements recorded by adjacent blue and yellow detectors.

**Figure 3 entropy-27-00868-f003:**
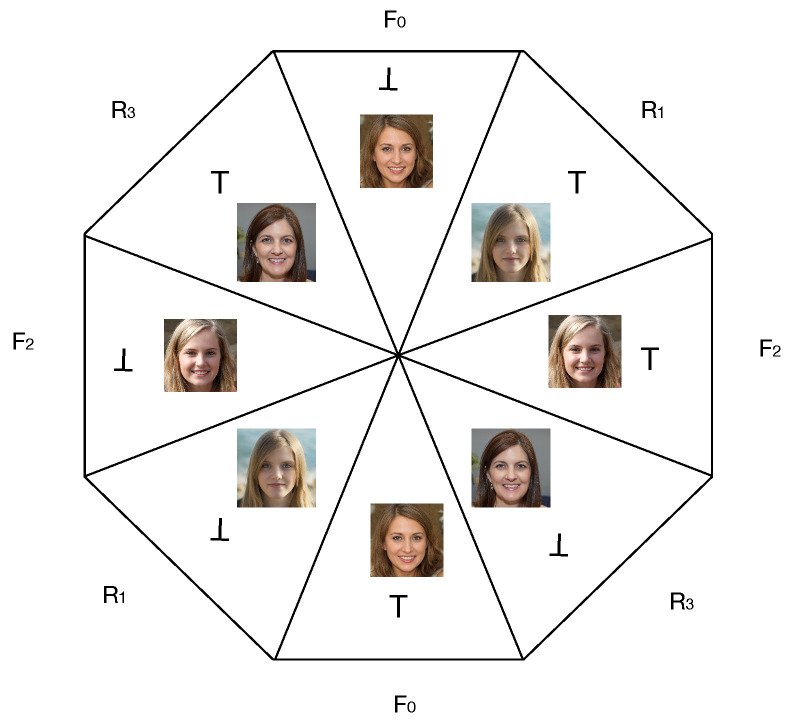
Schematic representation of a generalized coin n=4 used in the experiment, where each of the four facets of the coin is a real or fake face (instead of a color). There are four measurement contexts within which two faces are simultaneously judged, one real and one fake: $\left\{F_{0},R_{1}\right\},\left\{R_{1},F_{2}\right\},\left\{F_{2},R_{3}\right\},\left\{R_{3},F_{0}\right\}$. When tossed, two measurement outcomes can be observed per flip, where a measurement outcome is a judgement whether a face is fake (⊤) or not (⊥).

**Figure 4 entropy-27-00868-f004:**
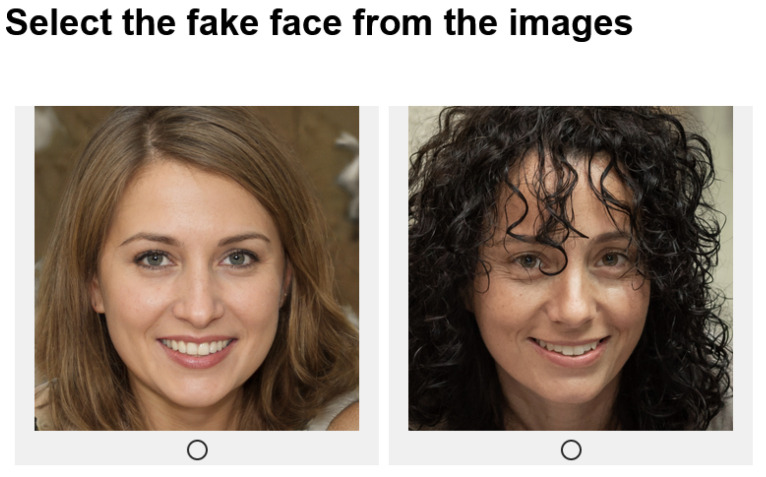
Example screenshot of the user interface for collecting human judgments.

**Table 1 entropy-27-00868-t001:** Context $C_{0}$.

$F_{0}$	$R_{1}$	*p*
⊥	⊥	0
⊤	⊥	$p_{0}$
⊥	⊤	$1-p_{0}$
⊤	⊤	0

Probability distribution from generalized coin flips in measurement context $C_{0}$.

**Table 2 entropy-27-00868-t002:** Context $C_{1}$.

$R_{1}$	$F_{2}$	*p*
⊥	⊥	0
⊤	⊥	$1-p_{1}$
⊥	⊤	$p_{1}$
⊤	⊤	0

Probability distribution from generalized coin flips in measurement context $C_{1}$.

**Table 3 entropy-27-00868-t003:** Context $C_{2}$.

$F_{2}$	$R_{3}$	*p*
⊥	⊥	0
⊤	⊥	$p_{2}$
⊥	⊤	$1-p_{2}$
⊤	⊤	0

Probability distribution from generalized coin flips in measurement context $C_{2}$.

**Table 4 entropy-27-00868-t004:** Context $C_{3}$.

$R_{3}$	$F_{0}$	*p*
⊥	⊥	0
⊤	⊥	$1-p_{3}$
⊥	⊤	$p_{3}$
⊤	⊤	0

Probability distribution from generalized coin flips in measurement context $C_{3}$.

**Table 5 entropy-27-00868-t005:** Contextuality analysis of the 20 coins n=4. CbD (bolded) signifies the coin is contextual.

Coin	$p_{0}$	$p_{1}$	$p_{2}$	$p_{3}$	Gender	Δ	CbD	Interval
1	0.56	0.46	0.42	0.46	F	0.28	**0.72**	[0.24,0.88]
2	0.62	0.72	0.60	0.60	F	0.24	**0.76**	[0.24,0.88]
3	0.32	0.30	0.78	0.72	F	0.96	0.04	[−0.32, 0.26]
4	0.32	0.78	0.36	0.12	F	1.32	−0.32	[−0.60,−0.02]
5	0.76	0.80	0.38	0.20	F	1.20	−0.20	[−0.52,0.06]
6	0.64	0.54	0.36	0.36	F	0.56	**0.44**	[0.02,0.66]
7	0.66	0.30	0.38	0.64	F	0.72	0.28	[−0.12,0.52]
8	0.76	0.84	0.55	0.66	F	0.58	**0.42**	[0.06,0.70]
9	0.28	0.58	0.52	0.30	F	0.60	0.40	[0, 0.62]
10	0.50	0.66	0.30	0.22	F	0.88	0.12	[−0.22,0.42]
11	0.80	0.62	0.28	0.22	M	1.16	−0.16	[−0.48,0.12]
12	0.72	0.74	0.50	0.54	M	0.48	**0.52**	[0.10,0.72]
13	0.16	0.34	0.26	0.18	M	0.36	**0.64**	[0.26,0.82]
14	0.50	0.32	0.84	0.66	M	1.04	−0.04	[−0.38,0.28]
15	0.42	0.48	0.56	0.44	M	0.26	**0.74**	[0.24,0.88]
16	0.48	0.62	0.36	0.36	M	0.52	**0.48**	[0.04,0.72]
17	0.50	0.26	0.24	0.42	M	0.52	**0.48**	[0.08,0.70]
18	0.40	0.70	0.54	0.14	M	1.12	−0.12	[−0.44,0.20]
19	0.40	0.38	0.76	0.88	M	1.00	0	[−0.43,0.22]
20	0.18	0.22	0.52	0.50	M	0.68	0.32	[−0.06,0.52]

**Table 6 entropy-27-00868-t006:** Averages for the Fake ($p_{0}$, $p_{2}$) and Real ($p_{1}$, $p_{3}$) face images, for female faces, male faces, and over all images.

	Fake	Real
Mean Female	0.50	0.51
Mean Male	0.47	0.45
Total Mean	0.49	0.48
Grand Mean	0.48	

## Data Availability

The original data presented in the study are openly available in https://github.com/pbruza/Contextuality_Real_Fake_faces (accessed on 10 August 2025).

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
