# Peer review of "Are Human Judgments of Real and Fake Faces Quantum-like Contextual?"

_entropy, 2025, doi:10.3390/e27080868_

Round 1

Reviewer 1 Report

Comments and Suggestions for Authors

Thank you for the opportunity to review your manuscript. I found the paper highly engaging. It applies Contextuality-by-Default to a timely and relevant topic: the comparison between fake and real images.

I recommend accepting the manuscript, subject to a few minor revisions.

COMMENTS

  1. I suggest clarifying in both the Introduction and Conclusion—using a few accessible and discursive sentences—what it means for judgments to be quantum-like contextual. This would benefit readers who are interested in the topic but may not be well-versed in technical details.
  2. If appropriate, I recommend briefly distinguishing quantum contextuality from order effects in experimental settings. In particular, it would be helpful to dialectically mention the Quantum Question Order (QQ) model, as discussed in: Busemeyer, J.R. & Bruza, P. Quantum Models of Cognition and Decision: Principles and Applications. Cambridge University Press.
  3. I suggest clarifying using a few accessible and discursive sentences—especially in the Discussion and/or Conclusion section—how acausality differs from causality, as this fundamental distinction may be of interest to non-expert readers.

Author Response

Reviewer 1:

Thank-you for your positive comments:

  1. I suggest clarifying in both the Introduction and Conclusion—using a few accessible and discursive sentences—what it means for judgments to be quantum-like contextual. This would benefit readers who are interested in the topic but may not be well-versed in technical details.

Response: Concise clarifications [in red] about what quantum-like contextuality means has been added to the Introduction and Conclusion sections

  1. If appropriate, I recommend briefly distinguishing quantum contextuality from order effects in experimental settings. In particular, it would be helpful to dialectically mention the Quantum Question Order (QQ) model, as discussed in: Busemeyer, J.R. & Bruza, P. Quantum Models of Cognition and Decision: Principles and Applications. Cambridge University Press.

Response: A distinction between contextuality and order effects has been added, including the suggested reference. See text [in red] on page 5.

  1. I suggest clarifying using a few accessible and discursive sentences—especially in the Discussion and/or Conclusion section—how acausality differs from causality, as this fundamental distinction may be of interest to non-expert readers.

Response: The difference between causality and acausality has been added in relation to the concept of context-sensitivity. (See text [in red] added to the Discussion section, page 9). The distinction is also repeated in the Conclusion.

Reviewer 2 Report

Comments and Suggestions for Authors

This is an instructive and significant contribution, which strongly suggests the quantum-like rather than (as some argue) possible classical-like nature of contextually of the phenomena considered, in conjunction with their acausal nature.  I would instead see it in terms of quantum probabilistic causality rather than lack of causality. Accordingly, my suggestion is that the author might consider the concept of quantum causality developed along related but sometimes different lines during the last decade of so. Also, it seems to me that more precision is necessary in speaking of Hilbert space operators (including vs. state vector) as representing any physical properties, even in realist interpretations, where they are still related to probabilities, via Born's rule, which is missing in the article as well. 

Author Response

Reviewer 2:

Thank-you for your positive comments.

  1. I would instead see it in terms of quantum probabilistic causality rather than lack of causality. Accordingly, my suggestion is that the author might consider the concept of quantum causality developed along related but sometimes different lines during the last decade of so.

Response: We draw on Pitowsky’s views, who gives up on causality when George Boole’s “conditions of Possible experience” are violated. These violations  are analogous  to the CbD inequality used in the paper (Equation 4). Therefore, we deem it to be justified to put forward acausal context-sensitivity to interpret the quantum-like contextuality exhibited by some generalized coins.

We have added a brief reference to quantum causality as an alternate view in the Discussion section. See text [in blue] on page 9.

  1. Also, it seems to me that more precision is necessary in speaking of Hilbert space operators (including vs. state vector) as representing any physical properties, even in realist interpretations, where they are still related to probabilities, via Born's rule, which is missing in the article as well. 

Response: The only place that Hilbert spaces are mentioned are on the bottom of page 3. The way the connection between Hilbert spaces and the KS theorem is stated is due to Tezzin, which we now explicitly acknowledge (page 3, line 97, text [in blue]). The you raise is valid. However, the generalized coin [refs 3, 4] is formalized using standard probability theory. We feel that introducing the corresponding Hilbert space representation is beyond the scope of the paper.

Reviewer 3 Report

Comments and Suggestions for Authors

I very much enjoyed reading this article. It is well written, an provides a valuable example to illustrate the presence of Type 2 Contextuality (Dzhafarov's true contextuality) in a classical setting. It also demonstrates that the contextuality of Tezzin's generalized coin can be obtained in a real experiment. Thus I hope that my comments while be taken in the spirit offered, to try to improve the quality of the paper overall.

In their Introduction, it seemed to me that the author's draw too heavily upon interpretations of contextuality derived from the physics literature. Lines 26-28 for example in which they say that context sensitivity cannot be reduced to traditional causal or probabilistic explanations seems rather strong and misleading. In entanglement and the generalized coin example as well I think, correlations are built into the very definition of the system under consideration, and require specific measurement arrangements for their demonstration. What mystifies in entanglement is that when the measurements of each observer are viewed independent of the other, one obtains marginal probabilities that are the same as for an independent entity (consistent connectedness according to Dzhafarov). It is only when the data is properly analyzed, treating the two component entities as a single entity, that the correlations are observed. In entanglement, the probabilities are correlated in the very definition of the entangled state. Confusion and mystery arise as a consequence of the confusion entailed by the myriad of interpretations of the wave function, which confuse ontology and epistemology. In the generalized coin, correlations are built into the structure of the coin, and measurements must be carried out in specific ways in order to reveal those correlations. Two observers examining only one feature of the coin would get results similar to a usual coin but if they observe suitably chosen regions of the coin, and then compare their results, they will observe these correlations. Confusion again arises from confusing marginal probabilities with etiologies - marginals that look alike need not have similar causes, and seemingly independent observations need not be independent when arising from a common cause, or a single entity.

There is nothing mysterious in the generalized coin, it is classical - the problem is that quantum physicists have made contextuality all about themselves and have projected their confusions onto every situation. Moreover, Dzhafarov shows that this has a Kolmogorovian explanation once the contextuality of the situation is properly taken into account - the existence of joint probabilities even in classical probability is not a given, but often an assumption, and this is only true when specific conditions are met, such as those of Vorob'ev or Boole. Contextuality, whether Type 1 or 2, is everywhere in the biological, psychological and social worlds. Just not in classical physics (the study of dead stuff).

Whether or not an entity has pre-existing values for a measurement depends upon exactly what is being measured and how those measurements are carried out. Many quantum mechanical arguments depend upon the specific formalism used (Hilbert spaces and self-adjoint operators) to calculate probabilities, but context dependent arguments can take place perfectly well within the usual Banach spaces with the addition of context labeling - again we get confused if we make the wrong assumptions about our system.

For example, Nagasawa showed in a deep argument, that the linear Schrodinger equation is equivalent to a nonlinear classical diffusion equation, both having exactly the same associated probability distribution given by the Born rule. So a perfectly classical model can be had if one gives up one assumption (here linearity).

Causation in psychology is also an enormously complex subject, requiring consideration of multiple levels of systems (see the work of Walter Freeman on mass action int he nervous system for example, in some of which notions of locality do not necessarily make any conceptual or mathematical sense. So we must be careful about bringing analogies from physics into areas where the core constructs might not even exist.

I wonder if the authors might soften some of their statements in their introduction.

I am also not sure that the conclusions ore supported by the evidence here. They show nicely that the generalized coin with the constraints upon measurements as given does indeed exhibit Type 2 contextuality. They then realize this experiment using human subjects essentially as coin flippers. They show that the failure of subjects to distinguish easily between real and fake (which is not necessarily an attribute of the image per se but rather is an attribute of its history or provenance) allows these subjects to produce a fair flipping of the "coin" as it were. It also demonstrates how scary this image fakery is becoming. The contextuality here lies not in the subjects necessarily, but is implicit in the test structure, in the structural organization of the image collection. The human observers are able to demonstrate this just as quantum mechanical observers demonstrate it of the entangled system.

Author Response

Reviewer 3.

We thank you for your positive comments. Some of the comments came across as informative background context, and thus it was sometimes challenging to discern what improvements you were advocating. We have responded as best we can in the following:

  1. In their Introduction, it seemed to me that the author's draw too heavily upon interpretations of contextuality derived from the physics literature. Lines 26-28 for example in which they say that context sensitivity cannot be reduced to traditional causal or probabilistic explanations seems rather strong and misleading.

Response: We have softened this claim. See text [in green] on page 1

  1. In entanglement and the generalized coin example as well I think, correlations are built into the very definition of the system under consideration, and require specific measurement arrangements for their demonstration. What mystifies in entanglement is that when the measurements of each observer are viewed independent of the other, one obtains marginal probabilities that are the same as for an independent entity (consistent connectedness according to Dzhafarov). It is only when the data is properly analyzed, treating the two component entities as a single entity, that the correlations are observed. In entanglement, the probabilities are correlated in the very definition of the entangled state. Confusion and mystery arise as a consequence of the confusion entailed by the myriad of interpretations of the wave function, which confuse ontology and epistemology. In the generalized coin, correlations are built into the structure of the coin, and measurements must be carried out in specific ways in order to reveal those correlations. Two observers examining only one feature of the coin would get results similar to a usual coin but if they observe suitably chosen regions of the coin, and then compare their results, they will observe these correlations

Response: Thanks for the interesting an informative context. We don’t use entanglement in our arguments as this can be viewed as controversial in the field of Quantum Cognition. In addition, Tezzin [refs 3,4] does not use the concept of entanglement to define the generalized coin. Tezzin shows how the generalized coin is ‘contextual’. Therefore, we focus our attention on contextuality, rather than entanglement.

  1. Confusion again arises from confusing marginal probabilities with etiologies - marginals that look alike need not have similar causes, and seemingly independent observations need not be independent when arising from a common cause, or a single entity.

Response: Agreed. But when the marginals do “look alike”, the accepted view in Quantum Cognition (ref 11)  is that there is no “disturbance”. Disturbance is usually attributed with having a causal basis. Therefore, lack of (sufficient) disturbance can be taken as acausality. We follow this approach.

  1. There is nothing mysterious in the generalized coin, it is classical - the problem is that quantum physicists have made contextuality all about themselves and have projected their confusions onto every situation. Moreover, Dzhafarov shows that this has a Kolmogorovian explanation once the contextuality of the situation is properly taken into account - the existence of joint probabilities even in classical probability is not a given, but often an assumption, and this is only true when specific conditions are met, such as those of Vorob'ev or Boole. Contextuality, whether Type 1 or 2, is everywhere in the biological, psychological and social worlds. Just not in classical physics (the study of dead stuff).

Response: Agreed. Hopefully our paper is a convincing illustration in the “psychological world”.

  1. Whether or not an entity has pre-existing values for a measurement depends upon exactly what is being measured and how those measurements are carried out.

Response: We agree with the following caveat. We follow Pitowsky and hold the view that if the coin is contextual, there is reason to believe that some cognitive properties of the participant are indeterminate, i.e., don’t have pre-existing values. If non-contextuality is present there is reason to believe that the properties have well-established pre-existing values. (See (Bruza et al., 2023)  for a more detailed argument).

We have added some text [in green] to contextualize your point in the context of our study (see lines 282-283 on page 9)

  1. Many quantum mechanical arguments depend upon the specific formalism used (Hilbert spaces and self-adjoint operators) to calculate probabilities, but context dependent arguments can take place perfectly well within the usual Banach spaces with the addition of context labeling - again we get confused if we make the wrong assumptions about our system

Response: We take this as an informative comment with no required change

  1. For example, Nagasawa showed in a deep argument, that the linear Schrodinger equation is equivalent to a nonlinear classical diffusion equation, both having exactly the same associated probability distribution given by the Born rule. So a perfectly classical model can be had if one gives up one assumption (here linearity).

Response: Thanks for this contrast, of which we were unaware. We assume no change was required.

  1. Causation in psychology is also an enormously complex subject, requiring consideration of multiple levels of systems (see the work of Walter Freeman on mass action int he nervous system for example, in some of which notions of locality do not necessarily make any conceptual or mathematical sense. So we must be careful about bringing analogies from physics into areas where the core constructs might not even exist.

Response: Agreed. The field of quantum cognition, of which this article is part, is very aware of this challenge. Causality is a concept both in physics and psychology and probabilistic analysis can allow some conclusions to be drawn. We have added extra clarification of how causality plays out in our study. (See text [in red] in the Discussion section, page 9)

  1. I wonder if the authors might soften some of their statements in their introduction.

Response: The claims around causality have been softened. See text [red and green] in the Introduction (page 1)

  1. I am also not sure that the conclusions ore supported by the evidence here. They show nicely that the generalized coin with the constraints upon measurements as given does indeed exhibit Type 2 contextuality. They then realize this experiment using human subjects essentially as coin flippers. They show that the failure of subjects to distinguish easily between real and fake (which is not necessarily an attribute of the image per se but rather is an attribute of its history or provenance) allows these subjects to produce a fair flipping of the "coin" as it were. It also demonstrates how scary this image fakery is becoming. The contextuality here lies not in the subjects necessarily, but is implicit in the test structure, in the structural organization of the image collection. The human observers are able to demonstrate this just as quantum mechanical observers demonstrate it of the entangled system.

Response: These are thought-provoking and informative remarks, particularly where the contextuality “lies”. We have added some text [in green, page 9, lines 284-285] that align with your remarks on this point. 

Round 2

Reviewer 3 Report

Comments and Suggestions for Authors

Acceptable